# Impact of Aortic Valve Regurgitation on Doppler Echocardiographic Parameters in Patients with Severe Aortic Valve Stenosis

**DOI:** 10.3390/diagnostics13111828

**Published:** 2023-05-23

**Authors:** Joscha Kandels, Michael Metze, Andreas Hagendorff, Stephan Stöbe

**Affiliations:** Klinik und Poliklinik für Kardiologie, Universitätsklinikum Leipzig, Liebigstr. 20, 04103 Leipzig, Germany

**Keywords:** transthoracic echocardiography, aortic stenosis, aortic regurgitation, continuity equation, 3D transesophageal echocardiography, doppler echocardiography

## Abstract

Background: Diagnosing severe aortic stenosis (AS) depends on flow and pressure conditions. It is suspected that concomitant aortic regurgitation (AR) has an impact on the assessment of AS severity. The aim of this study was to analyze the impact of concomitant AR on Doppler-derived guideline criteria. We hypothesized that both transvalvular flow velocity (maxV_AV_) and the mean pressure gradient (mPG_AV_) will be affected by AR, whereas the effective orifice area (EOA) and the ratio between maximum velocity of the left ventricular outflow tract and transvalvular flow velocity (maxV_LVOT_/maxV_AV_) will not. Furthermore, we hypothesized that EOA (by continuity equation), and the geometric orifice area (GOA) (by planimetry using 3D transesophageal echocardiography, TEE), will not be affected by AR. Methods and Results: In this retrospective study, 335 patients (mean age 75.9 ± 9.8 years, 44% male) with severe AS (defined by EOA < 1.0 cm^2^) who underwent a transthoracic and transesophageal echocardiography were analyzed. Patients with a reduced left ventricular ejection fraction (LVEF < 53%) were excluded (*n* = 97). The remaining 238 patients were divided into four subgroups depending on AR severity, and they were assessed using pressure half time (PHT) method: no, trace, mild (PHT 500–750 ms), and moderate AR (PHT 250–500 ms). maxV_AV_, mPG_AV_ and maxV_LVOT_/maxV_AV_ were assessed in all subgroups. Among the four subgroups (no (*n* = 101), trace (*n* = 49), mild (*n* = 61) and moderate AR (*n* = 27)), no differences were obtained for EOA (no AR: 0.75 cm^2^ ± 0.15; trace AR: 0.74 cm^2^ ± 0.14; mild AR: 0.75 cm^2^ ± 0.14; moderate AR: 0.75 cm^2^ ± 0.15, *p* = 0.998) and GOA (no AR: 0.78 cm^2^ ± 0.20; trace AR: 0.79 cm^2^ ± 0.15; mild AR: 0.82 cm^2^ ± 0.19; moderate AR: 0.83 cm^2^ ± 0.14, *p* = 0.424). In severe AS with moderate AR, compared with patients without AR, maxV_AV_ (*p* = 0.005) and mPG_AV_ (*p* = 0.022) were higher, whereas EOA (*p* = 0.998) and maxV_LVOT_/maxV_AV_ (*p* = 0.243) did not differ. The EOA was smaller than the GOA in AS patients with trace (0.74 cm^2^ ± 0.14 vs. 0.79 cm^2^ ± 0.15, *p* = 0.024), mild (0.75 cm^2^ ± 0.14 vs. 0.82 cm^2^ ± 0.19, *p* = 0.021), and moderate AR (0.75 cm^2^ ± 0.15 vs. 0.83 cm^2^ ± 0.14, *p* = 0.024). In 40 (17%) patients with severe AS, according to an EOA < 1.0 cm^2^, the GOA was ≥ 1.0 cm^2^. Conclusion: In severe AS with moderate AR, the maxV_AV_ and mPG_AV_ are significantly affected by AR, whereas the EOA and maxV_LVOT_/maxV_AV_ are not. These results highlight the potential risk of overestimating AS severity in combined aortic valve disease by only assessing transvalvular flow velocity and the mean pressure gradient. Furthermore, in cases of borderline EOA, of approximately 1.0 cm^2^, AS severity should be verified by determining the GOA.

## 1. Introduction

The prevalence of severe aortic valve stenosis (AS) as the most common type of valvular heart disease increases with age (>75 years) to 3–4% [1]. Although the prevalence of aortic regurgitation (AR) is about 5%, it increases up to 75% in patients with AS [2]. Recent recommendations for AS evaluation are primarily based on Doppler-derived parameters by transthoracic echocardiography (TTE) [3]. Peak transvalvular flow velocity (maxV_AV_), mean transvalvular pressure gradient (mPG_AV_), the ratio between the maximum pre-stenotic velocity in the left ventricular outflow tract (maxV_LVOT_) and maxV_AV_ (maxV_LVOT_/maxV_AV_), and the effective orifice area (EOA) that is calculated using the continuity equation, are recommended parameters for grading AS severity [3]. Nevertheless, the EOA or geometric orifice area (GOA) represents the actual target value for the assessment of AS severity, and thus, it forms the basis for further divisions into hemodynamic AS subgroups [4,5].

The EOA describes the functional orifice area, which can be indirectly calculated using the continuity equation. When determining the EOA, some methodological limitations must be considered. For example, in comparison to the GOA, the EOA is suspected to be smaller, which is due to the phenomenon of flow constriction caused by flow turbulence at the inner edges of the actual anatomic orifice area [6]. In contrast, the GOA corresponds with the anatomic orifice area, which can directly be measured by echocardiography. Semi-invasive transesophageal (3D) echocardiography (TEE) enables a more precise assessment of the GOA and aortic dimensions. TEE is also recommended when components of the continuity equation (e.g., the left ventricular outflow tract (LVOT) diameter) cannot be determined precisely using TTE. Furthermore, LV volumes and function, as well as aortic dimensions, can be determined more precisely by 3D echocardiography [7,8,9].

Most studies on clinical outcomes reported on patients with isolated AS [5,10]. The outcome data of patients with severe AS and concomitant AR are scarce, although a few studies suggested a similar or unfavorable outcome for patients with combined aortic valve disease [11,12,13,14]. The increase of forward blood flow through the aortic valve in the presence of AR will not influence the calculation of EOA; however, the flow conditions, which are estimated by assessing the forward stroke volume through the aortic valve, and which are determined by Doppler echocardiography, do not reflect effective stroke volume [15]. This difference between total and effective stroke volume in mixed aortic valve disease might influence the classification of patients with different AS subgroups.

The aim of this study was to analyze the impact of concomitant AR on Doppler-derived parameters (maxV_AV_, mPG_AV_, EOA, maxV_LVOT_/maxV_AV_) in patients with severe AS, and to detect potential discrepancies between the EOA (by continuity equation) and GOA (by planimetry). Regarding current recommendations concerning grading AS severity, and the increase of total forward flow due to AR, we hypothesized that concomitant AR has no significant impact on the Doppler-derived guideline criteria of severe AS (EOA and maxV_LVOT_/maxV_AV_), but it does have a significant impact on maxV_AV_ and mPG_AV_.

## 2. Methods

This retrospective study included 335 patients with severe AS (defined by EOA < 1 cm^2^ and indexed EOA < 0.6 cm^2^/m^2^) who underwent TTE and TEE at University Hospital Leipzig (Leipzig, Germany) between January 2014 and December 2017. Patients with a reduced left ventricular ejection fraction (LVEF) < 55% (*n* = 97) were excluded. The remaining 238 patients were divided into four subgroups, depending on AR severity, and this was determined by the pressure half time (PHT) method: no, trace, mild (PHT 500–750 ms), and moderate AR (PHT 250–500 ms) (Figure 1). Severe AS patients with severe AR (PHT < 250 ms) were not observed. Patients’ symptoms were collected from medical records.

TTE was performed using a Vivid e9 or Vivid e95 ultrasound system with a M5-S phased array probe (GE Healthcare Vingmed Ultrasound AS, Horten, Norway). Data analyses were performed with the EchoPac software (Version 202, GE Healthcare Vingmed Ultrasound AS, Horten, Norway).

### 2.1. Morphology and Function

Left ventricular dimensions and LV mass (LVM) were assessed using M-Mode measurements that were obtained using the Devereux formula. Concentric LV hypertrophy was defined by the relative wall thickness:  >  0.42 and LVM index > 95 g/m^2^ (female) or >115 g/m^2^ (male) [16]. The LV volumes and LVEF were assessed with biplane LV planimetry using the modified Simpson’s rule. LVSV was also assessed by Doppler echocardiography (LVSVi_Dop_). All measurements were performed in accordance with current recommendations [16,17]. Left ventricular global longitudinal strain (GLS) was measured using 2D speckle tracking analysis along the apical long axis-, 2-, and 4-chamber, in accordance with current recommendations [18,19]. Endocardial contours and tracking areas were adjusted manually to enable full myocardial tracking. Only segments with accurate tracking were accepted. Diastolic function was assessed using E/E’, E/A, and systolic pulmonary artery pressure (sPAP), in accordance with current recommendations [20].

### 2.2. Aortic Valve Stenosis

In all patients, the echocardiographic parameters recommended for AS evaluation were determined: maxV_AV_, mPG_AV_, and the maxV_LVOT_/maxV_AV_ ratio. The effective orifice area (EOA) was calculated using the continuity equation [3,4]. The diameter of the left ventricular outflow tract (D_LVOT_) was determined using TEE along the mid-esophageal long axis at 5–10 mm from the aortic valve during mid-systole. LVOT blood flow velocities were assessed using pulsed wave (pw)-Doppler along the apical long axis, wherein the sample volume was placed at a position that corresponded with the position of the D_LVOT_ measurements. maxV_AV_ was determined using continuous wave (cw)-Doppler along the apical long axis. mPG_AV_ was calculated using the Bernoulli equation [3,4]. The maxV_LVOT_/maxV_AV_ ratio was calculated using the maximum flow velocities of the LVOT and the aortic valve.

### 2.3. Aortic Valve Regurgitation

The pressure half time (PHT) was assessed along the apical long axis in diastole using cw-Doppler after the optimal visualization of the regurgitation jet. Trace AR was defined by the following criteria: (1) a pinhead-sized origin of the regurgitation jet; (2) a PHT > 750 ms, if cw-Doppler documented no intercept angle between the ultrasound beam and the direction of the blood flow of the regurgitant velocities; and/or (3) a non-holodiastolic AR documented by an anatomical color-M-Mode. A PHT of 500–750 ms was defined as mild AR, and a PHT of 250–499 ms was defined as moderate AR.

### 2.4. Classification by Flow and Pressure Gradients

In accordance with the proposal of Lancellotti et al. [5], all patients were classified in accordance with their flow and pressure conditions. mPG_AV_ < 40 mmHg was defined as a low gradient (LG)-AS, and mPG_AV_ ≥ 40 mmHg as a high gradient (HG)-AS. A left ventricular stroke volume index (LVSVi_Dop_) ≤ 35 mL/m^2^ was defined as a low flow (LF)-AS, and LVSVi_Dop_ > 35 mL/m^2^ as a normal flow (NF)-AS. Subsequently, severe AS patients were divided into four subgroups: LFLG (low flow-low gradient)-AS, NFLG (normal flow-low gradient)-AS, LFHG (low flow-high gradient)-AS, and NFHG (normal flow-high gradient)-AS.

### 2.5. Transesophageal Echocardiography

TEE was performed using a Vivid e9 or Vivid e95 ultrasound system with a 6VT transesophageal array probe (GE Healthcare Vingmed Ultrasound AS, Horten, Norway). The geometric orifice area was determined using planimetry with 3D echocardiography. During the post-processing analyses, sectional planes were manually aligned in the plane of the aortic valve opening, and they were orthogonally parallel to the course of the ascending aorta (“flexi-slice mode”) in order to best visualize the aortic valve opening.

### 2.6. Statistical Analysis

Statistical analyses were performed using SPSS Statistics version 24.0 (IBM, Armonk, NY, USA). Continuous variables were expressed as the mean value ± standard deviation (SD), and they were compared between groups using the Student’s *t*-test. All categorical variables were expressed as numbers with their percentages (%) and compared using the chi-squared or Fisher exact test, as appropriate. The Kolmogorov–Smirnov test was performed to test the normal distribution of the population. Linear regression and Pearson’s r were applied to evaluate the association between two linear variables. Data comparisons between more than two groups were performed by one-way Analysis of Variance (ANOVA). A *p* value < 0.05 was considered to indicate statistical significance.

Intraobserver variability was assessed by repeating all measurements under the same conditions in 20 patients. Furthermore, interobserver variability was assessed using the measurements of a second investigator who was unaware of the results of the first examination.

## 3. Results

In this retrospective study, 238 patients (44% male, mean age 75.9 ± 9.8 years) showed an EOA < 1 cm^2^ (indexed EOA < 0.6 cm^2^/m^2^). In 42% (*n* = 101) of AS patients, no AR was detected, whereas 21% (*n* = 49) showed trace, 26% (*n* = 61) mild, and 11% (*n* = 27) moderate AR. Baseline characteristics are presented in Table 1.

### 3.1. Basic Echocardiographic Parameters

Left ventricular dimensions and EF did not differ significantly between AR subgroups (Table 2 and Table 3). In 97% of AS patients, concentric LV hypertrophy was observed. The LV stroke volume index was determined using biplane planimetry (LVSVi_BP_), and it was significantly higher in patients with moderate AR (Table 2). There were no significant differences of E/E’ and E/A among all subgroups, whereas the sPAP was significantly higher in patients with moderate AR (Table 3). Global longitudinal strain was lowest in AS patients with moderate AR without reaching statistical significance (Table 3).

### 3.2. Doppler-Derived Echocardiographic Parameters

The lowest maxV_AV_ and mPG_AV_ were observed in patients without AR, whereas the highest maxV_AV_ and mPG_AV_ were found in patients with moderate AR (Table 4). Peak transvalvular flow velocities of the aortic valve were higher in patients with moderate AR compared with those with no (*p* = 0.005), trace (*p* = 0.015), and mild AR (*p* = 0.044). Accordingly, mPG_AV_ was significantly higher in patients with moderate AR compared with AS patients with no (*p* = 0.022) and trace AR (*p* = 0.028). The maxV_LVOT_/maxV_AV_ ratio did not differ between AR subgroups (*p* = 0.243).

### 3.3. Determination ofAaortic Valve Area

The effective (EOA) and geometric (GOA) orifice area did not differ significantly between AR subgroups (Table 4). In general, the EOA was lower than GOA in all patients with trace (0.74 cm^2^ ± 0.14 vs. 0.79 cm^2^ ± 0.15, *p* = 0.024), mild (0.75 cm^2^ ± 0.14 vs. 0.82 cm^2^ ± 0.19, *p* = 0.006), and moderate AR (0.75 cm^2^ ± 0.15 vs. 0.83 cm^2^ ± 0.14, *p* = 0.006). No significant differences between EOA and GOA could be observed in patients without AR (0.75 cm^2^ ± 0.15 vs. 0.78 cm^2^ ± 0.20, *p* = 0.135).

### 3.4. Classification by Flow and Pressure Gradients

In AS patients with moderate AR, a lower number of patients with LFLG conditions (15% vs. 38%; *p* = 0.025), and a higher number of patients with NFHG conditions (41% vs. 22%; *p* = 0.046), was observed compared with AS patients without AR (Figure 2). The number of AS patients with LFHG conditions and NFLG conditions did not differ significantly between AR subgroups (*p* > 0.05) (Figure 2).

### 3.5. Symptoms

Among AS patients with concomitant AR, dyspnea was the most common symptom, followed by chest pain and syncope (Figure 3). The number of symptoms did not differ between AR subgroups (*p* > 0.05).

### 3.6. Intra- and Interobserver Variability

Intra- and interobserver variabilities of all echocardiographic measurements were in the range of 7.2% to 8.6%.

## 4. Discussion

### The Main Findings of the Present Study

Doppler-derived parameters, maxV_AV_ and mPG_AV_, were significantly increased in severe AS and concomitant moderate AR, thus confirming the hypothesis that AR has a significant impact on these echocardiographic parameters.

The EOA calculated using the continuity equation and maxV_LVOT_/maxV_AV_, did not differ significantly between AR subgroups. In addition, the hypothesis concerning the idea that in AS patients with combined AR, EOA is significantly lower than GOA, was confirmed. As suspected, there was no significant difference between either parameter in patients with isolated severe AS.

In patients with moderate AR, the number of patients with LFLG conditions was significantly lower than in patients without AR, whereas the number of patients with NFHG conditions was significantly higher in patients with moderate AR compared with patients without AR.

In this retrospective study, patients were comparatively old and had a low body height; this was associated with small LV cavities, and consequently, lower LV stroke volumes. These characteristics may explain that although all patients presented a LVEF > 55%, about one third (29%) presented LFLG conditions and fulfilled the criteria for “paradoxical” LFLG-AS, which was first described in 2007 by Hachicha et al. [21]. Higher age and lower body height have been previously described as possible explanations for a higher incidence of paradoxical LFLG-AS in severe AS [22]. Another aspect that needs to be considered are the smaller LV diameters and volumes due to concentric remodeling and LV hypertrophy; this was observed in almost all patients and these aspects are strongly associated with severe AS [23]. Even though LVEDV did not differ significantly, a slight increase in LVEDV with increasing trace to moderate AR, presumably due to the increasing regurgitant volume, was observed. The similar LVEDV values in all AS subgroups, regardless of AR severity, suggest that concentric LV hypertrophy caused by AS is not accompanied by LV dilation due to volume overload at the different stages of AR observed in this study. The unchanged LV geometry between subgroups is also reflected by similar values of diastolic function parameters such as E/E’.

In 1988, Grayburn et al. observed the significant impact of AR on doppler-derived parameters (maxV_AV_ and mPG_AV_) in AS patients in a small cohort of 25 patients [24]. This is in line with the observations of the present study and can be explained by the increased forward stroke volume that is superimposed by the regurgitant volume, thus contributing to an increase in maxV_AV_ and mPG_AV_. Although both the maximum velocities increase over the LV outflow tract and at the level of the aortic valve, the ratio between the two (maxV_LVOT_/maxV_AV_ ratio) remains the same. In contrast, mPG_AV_ increases due to the proportional regurgitant volume, thus leading to an overestimation of AS severity based on mPG_AV_. Therefore, maxV_AV_ and mPG_AV_ are less suitable compared with EOA and maxV_LVOT_/maxV_AV_, which must be considered when estimating AS severity in patients with concomitant AR.

In all patients with severe AS, irrespective of AR severity, the EOA was smaller than the GOA, which can be attributed to rheological principles. As the flow increases, the turbulences adjacent to the central jet formation increase, causing a predominant narrowing of the central aligned flow; this characterizes the EOA, as shown by in vitro data [25]. Thus, in AS with higher degree of AR severity and higher transvalvular flow velocities, the difference between EOA and GOA increased. Furthermore, this observation can be explained by the effect of AV calcification on the aortic annulus, which leads to an underestimation of the LVOT diameter, and thus, compared with the GOA, it leads to lower EOA values that are calculated using the continuity equation [26,27,28]. The difference between EOA and GOA grows as the proportion of the regurgitant volume increases, which indicates the importance of TEE in this specific cohort of patients. On the other hand, although the EOA corresponds with the functional impairment, the GOA only reflects the physical size of the valve opening, and it does not account for these functional aspects. The American College of Cardiology/American Heart Association (ACC/AHA) guidelines on the management of valvular heart disease recommend using the EOA. Nevertheless, according to our data, especially in patients with moderate to severe AS, TEE may give more detailed information.

Invasive measurements of pressure gradients to assess AS severity are possible in principle, but are only recommended in patients where non-invasive cardiac imaging is inconclusive or discordant with clinical findings. Nevertheless, in patients with severe AS and a high cardiovascular risk profile, coronary heart disease should be excluded with a coronary angiography before further treatments (e.g., transcatheter aortic valve implantation or surgery) are performed [3].

Lancellotti et al. demonstrated differences between clinical outcomes in patients with severe AS depending on the transvalvular pressure gradient and indexed forward stroke volume [5]. In accordance with the results of the present study, AR not only has a significant impact on maxV_AV_ and mPG_AV_, but also on the echocardiographic parameters of the proposed classification by Lancellotti et al. [5]. The more patients with NFHG conditions, such as patients with severe AS and concomitant moderate AR, can presumably be explained by an overestimation of the LV stroke volume index (LVSVi); this is measured using the Doppler method due to turbulences in the LVOT caused by an increase of forward blood flow due to AR. The simple application of the classification by Lancellotti et al. in patients with combined AV disease might distort the clinical outcome for these patients. It must be noted that only patients with asymptomatic isolated severe AS were included in the study by Lancellotti et al. [5], whereas both symptomatic and asymptomatic (only 18%) severe AS patients were included in the present study. Thus, it can be assumed that effective LVSV presumably enables a better classification of flow conditions in patients with combined AV disease.

The fact that symptoms did not differ between the four subgroups with severe AS, each with different degrees of AR severity, might lead to the assumption that trace to moderate AR is well compensated in severe AS.

Based on the results of the present study, the question arises as to whether the classification of AS severity based on Doppler-derived parameters (maxV_AV_, mPG_AV_, EOA, and maxV_LVOT_/maxV_AV_) is more error-prone if, for example, maxV_AV_ and mPG_AV_, are significantly influenced by the contribution of the concomitant AR. Thus, AS severity might be overestimated in patients with only moderate AS if the impact of the proportional regurgitant volume of the AR will not be considered.

On the other hand, would it be equally possible that AS severity will be underestimated if maxV_AV_ and mPG_AV_ are still within the normal ranges? This question should be answered by an assessment of EOA, provided that all methodological requirements have been considered. However, due to possible uncertainty regarding the classification of AS severity, the determination of the GOA using TEE should be included in the diagnostic algorithm [7].

## 5. Strengths and Limitations

In the present study, a relatively high number of well-characterized patients with severe AS (according to the results of a continuity equation) were included (*n* = 283). In addition, all patients underwent TEE to confirm the diagnosis. Low intra- and interobserver variabilities highlight the quality of the acquired and analyzed echocardiographic data sets. In addition to the echocardiographic data, data on the patients’ symptoms were also available.

Although the effective regurgitant orifice area (EROA), determined by the proximal isovelocity surface area (PISA) method, may be considered a more reliable parameter, AR severity was assessed semi-quantitatively using PHT in the present study. It should be noted that in patients with severe AS, and with only trace or mild AR (most patients in the present study), flow convergence zones (PISA zones) are difficult to visualize, and the PISA method is often not applicable. As this is a retrospective study, the flow convergence zones could not be well elaborated in many patients with primarily trace and mild AR. On the other hand, the PHT method is also mentioned in current guidelines, despite its main limitation that the alignment of AR jet formation using cw-Doppler is often difficult.

## 6. Conclusions

In severe AS, concomitant AR has a significant impact on the Doppler-derived guideline criteria, maxV_AV_ and mPG_AV_, whereas the EOA and the ratio of maxV_LVOT_/maxV_AV_ remain unchanged. Characterization of AS severity using only maxV_AV_ and mPG_AV_ might overestimate AS severity in patients with concomitant moderate AR. Further prospective studies need to clarify the prognostic impact of concomitant AR on AS severity—especially in borderline EOA—using the continuity equation. In addition, TEE should frequently be implemented in the diagnostic algorithm to confirm the diagnosis of severe AS.

## Figures and Tables

**Figure 1 diagnostics-13-01828-f001:**
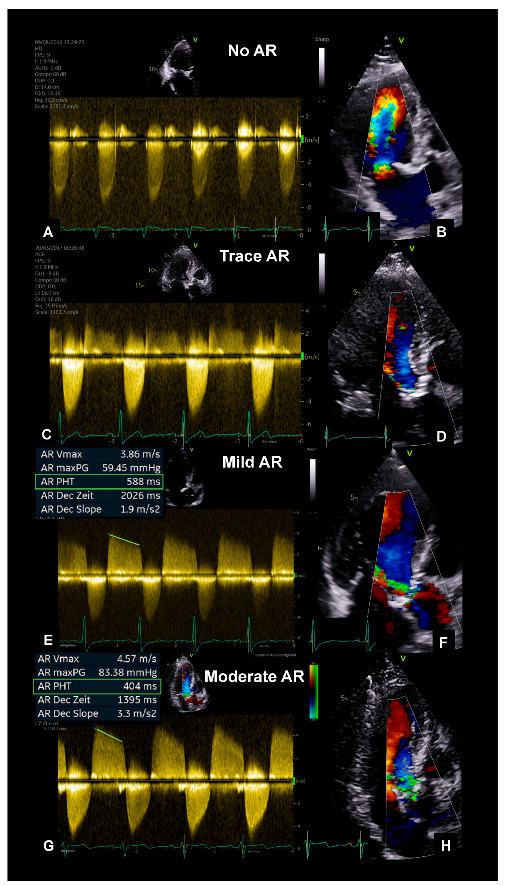
Assessment of AR in patients with severe AS. cw−Doppler spectrum and a color−coded−single frame of the apical long axis during end-diastole are illustrated for a patient with no (**A**,**B**), trace (**C**,**D**), mild (**E**,**F**), and moderate AR (**G**,**H**). Pressure half time (PHT) measured by cw−Doppler across the aortic valve along the apical long axis in mild and moderate AR (**E**,**G**). AR = Aortic regurgitation; AS = Aortic stenosis; cw = continuous wave.

**Figure 2 diagnostics-13-01828-f002:**
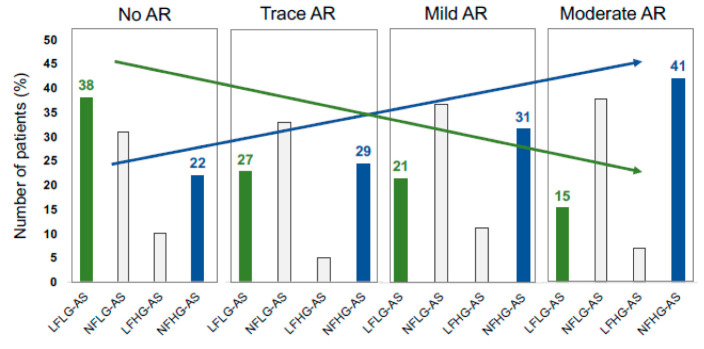
Changes in AS subtype frequencies with increasing aortic regurgitation (AR) severity. Classification of aortic stenosis (AS) according to Lancellotti et al. [5]. LF = Low flow; NF = Normal flow; LG = Low gradient; HG = High gradient.

**Figure 3 diagnostics-13-01828-f003:**
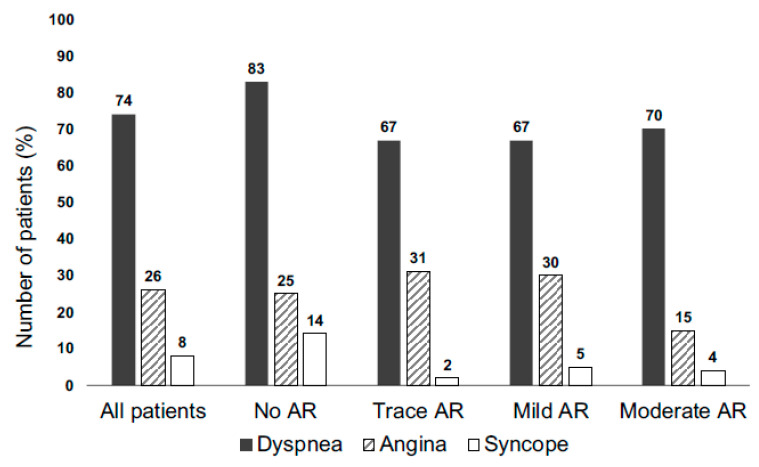
Prevalence of symptoms in mixed aortic valve disease. AR = Aortic regurgitation.

**Table 1 diagnostics-13-01828-t001:** Baseline characteristics.

Variables	All Patients (*n* = 238)	No AR (*n* = 101)	Trace AR (*n* = 49)	Mild AR (*n* = 61)	Moderate AR (*n* = 27)	*p* Value
Age (year)	75.9 ± 9.8	75.7 ± 9.4	75.4 ± 10.6	76.2 ± 10.1	76.7 ± 9.0	0.998
Sex (% male)	104 (44%)	42 (42%)	21 (43%)	30 (49%)	11 (41%)	0.667
Height (cm)	165.7 ± 9.5	165.7 ± 10.3	166.9 ± 9.6	165.9 ± 7.9	163.2 ± 9.4	0.445
Weight (kg)	76.3 ± 16.4	76.5 ± 18.7	78.6 ± 12.0	75.4 ± 15.8	73.2 ± 15.1	0.546
BSA (m^2^)	1.87 ± 0.23	1.86 ± 0.27	1.90 ± 0.18 ^⧧^	1.86 ± 0.21	1.81 ± 0.22	0.451
BMI (kg/m^2^)	27.7 ± 5.1	27.7 ± 5.6	28.3 ± 4.2	27.4 ± 5.2	27.4 ± 5.0	0.812
sBP (mmHg)	137.4 ± 21.5	136.8 ± 21.4	133.3 ± 21.2	141.3 ± 21.9	138.3 ± 21.2	0.272
dBP (mmHg)	80.0 ± 12.9	80.3 ± 11.7	79.5 ± 13.0	81.6 ± 14.7	76.2 ± 12.7	0.335
HR (1/min)	75.7 ± 16.1	79.0 ± 18.0 ^†,⧧^	74.6 ± 13.3	70.8 ± 13.8	76.2 ± 15.4	0.016 *

* significant difference (*p*  <  0.05) with trace AR group. ^†^ significant difference with mild AR group. ^⧧^ significant difference with moderate AR group; AR = aortic regurgitation; BSA = body surface area; BMI = body-mass index; sBP = systolic blood pressure; dBP = diastolic blood pressure; HR = heart rate.

**Table 2 diagnostics-13-01828-t002:** Parameters of left ventricular morphology and volumes.

Variables	All Patients (*n* = 238)	No AR (*n* = 101)	Trace AR (*n* = 49)	Mild AR (*n* = 61)	Moderate AR (*n* = 27)	*p* Value
IVSD (mm)	14.6 ± 2.8	14.5 ± 2.9	14.9 ± 2.4	14.5 ± 2.7	14.8 ± 3.2	0.824
LVPWD (mm)	14.5 ± 2.9	14.4 ± 2.6	14.2 ± 1.9	14.7 ± 4.0	14.9 ± 2.4	0.693
LVEDD (mm)	44.4 ± 6.1	43.7 ± 6.4 ^†^	44.2 ± 6.2	45.5 ± 6.1	44.7 ± 4.4	0.331
LVESD (mm)	28.2 ± 5.6	28.0 ± 6.1	28.2 ± 5.7	28.5 ± 5.1	28.0 ± 5.0	0.999
RWT	0.67 ± 0.18	0.67 ± 0.17	0.66 ± 0.15	0.66 ± 0.24	0.67 ± 0.12	0.999
LVMi (g/m^2^)	141.1 ± 39.7	137.2 ± 41.2	136.0 ± 30.9	147.3 ± 42.0	151.2 ± 45.0	0.178
LVEDV (mL)	105.6 ± 30.0	103.3 ± 30.8	104.3 ± 24.1	107.7 ± 33.1	111.5 ± 29.6	0.569
LVESV (mL)	36.2 ± 14.7	36.3 ± 14.0	35.7 ± 13.4	36.2 ± 16.7	37.2 ± 15.3	0.999
LVSV_BP_ (mL)	69.4 ± 18.7	67.0 ± 19.7 ^⧧^	68.6 ± 14.2	71.5 ± 19.8	74.4 ± 18.5	0.215
LVSVi_BP_ (mL)	36.9 ± 9.5	35.2 ± 10.0 ^†,⧧^	36.2 ± 7.7 ^⧧^	38.4 ± 9.3	41.1 ± 9.3	0.015 *
LVSV_Dop_ (mL)	69.4 ± 17.6	67.4 ± 17.4	69.5 ± 16.2	71.3 ± 19.2	72.8 ± 16.9	0.340
LVSVi_Dop_ (mL)	37.4 ± 9.0	36.3 ± 8.6 ^⧧^	36.7 ± 8.6 ^⧧^	38.4 ± 9.5	40.7 ± 9.8	0.105

* significant difference (*p*  <  0.05) with trace AR group. ^†^ significant difference with mild AR group. ^⧧^ significant difference with moderate AR group; AR = aortic regurgitation; IVSD = interventricular septum diameter; LVPWD = left ventricular posterior wall diameter; LVEDD = left ventricular end-diastolic diameter; LVESD = left ventricular end-systolic diameter; RWT = relative wall thickness; LVMi = left ventricular mass index; LVEDV = left ventricular end-diastolic volume; LVESV = left ventricular end-systolic volume; LVSV_i_ = left ventricular stroke volume index; BP = Biplane; Dop = Doppler method.

**Table 3 diagnostics-13-01828-t003:** Parameters of left ventricular systolic and diastolic function.

Variables	All Patients (*n* = 238)	No AR (*n* = 101)	Trace AR (*n* = 49)	Mild AR (*n* = 61)	Moderate AR (*n* = 27)	*p* Value
LVEF (%)	66.0 ± 7.1	65.2 ± 6.6	65.4 ± 6.7	66.6 ± 7.3	67.9 ± 8.4	0.259
GLS (%)	−15.5 ± 7.5	−15.0 ± 7.4	−15.1 ± 7.4	−16.7 ± 6.7	−14.5 ± 9.9	0.468
CI (L/m^2^)	2.5 ± 0.06	2.6 ± 0.8 ^†,⧧^	2.5 ± 0.6 ^⧧^	2.4 ± 0.7 ^⧧^	3.0 ± 0.7	0.004 *
E/E’	19.3 ± 9.5	17.8 ± 7.7 ^⧧^	17.8 ± 7.8	19.3 ± 8.4	21.1 ± 9.2	0.215
E/A	1.69 ± 1.85	1.69 ± 2.03 ^⧧^	1.69± 1.87	1.87 ± 2.1	1.25 ± 0.59	0.578
LAEDV (mL)	70.3 ± 29.2	68.3 ± 29.5 ^⧧^	65.9 ± 17.2 ^⧧^	73.3 ± 32.6	79.7 ± 30.5	0.154
sPAP (mmHg)	40.7 ± 12.1	39.8 ± 12.2 ^⧧^	39.2 ± 12.4 ^⧧^	40.5 ± 10.8 ^⧧^	47.6 ± 12.8	0.018 *

* significant difference (*p*  <  0.05) with trace AR group. ^†^ significant difference with mild AR group. ^⧧^ significant difference with moderate AR group; AR = aortic regurgitation; LVEF = left ventricular ejection fraction; GLS = global longitudinal strain; CI = cardiac index; LAEDV = left atrial end-diastolic volume; sPAP = systolic pulmonary artery pressure.

**Table 4 diagnostics-13-01828-t004:** Echocardiographic criteria for the assessment of aortic valve stenosis severity.

Variables	All Patients (*n* = 238)	No AR (*n* = 101)	Trace AR (*n* = 49)	Mild AR (*n* = 61)	Moderate AR (*n* = 27)	*p* Value
EOA (cm^2^)	0.75 ± 0.14	0.75 ± 0.15	0.74 ± 0.14	0.75 ± 0.14	0.75 ± 0.15	0.998
GOA (cm^2^)	0.80 ± 0.18	0.78 ± 0.20	0.79 ± 0.15	0.82 ± 0.19	0.83 ± 0.14	0.424
maxV_AV_ (m/s)	4.12 ± 0.77	4.04 ± 0.76 ^⧧^	4.08 ± 0.68 ^⧧^	4.15 ± 0.85 ^⧧^	4.45 ± 0.70	0.097
maxV_LVOT_(m/s)	0.98 ± 0.21	0.98 ± 0.22 ^⧧^	0.94 ± 0.17 ^⧧^	0.98 ± 0.24 ^⧧^	1.10 ± 0.19	0.019 *
mPG_AV_ (mmHg)	37.2 ± 15.5	35.6 ± 15.3 ^⧧^	35.7 ± 13.3 ^⧧^	38.7 ± 16.3	43.1 ± 16.8	0.109
maxV_LVOT_/maxV_AV_	0.24 ± 0.06	0.25 ± 0.06	0.24 ± 0.05	0.24 ± 0.07	0.25 ± 0.05	0.243

* significant difference (*p*  <  0.05) with trace AR group. ^⧧^ significant difference with moderate AR group; AR = aortic regurgitation; EOA = effective orifice area; AVA = aortic valve area; maxV_AV_ = maximum transvalvular flow velocity; maxV_LVOT_ = maximum flow velocity LVOT; mPG_AV_ = mean pressure gradient of aortic valve.

## Data Availability

The data sets used and/or analyzed during the current study are available from the corresponding author upon reasonable request.

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
