# Peer review of "Impact of Aortic Valve Regurgitation on Doppler Echocardiographic Parameters in Patients with Severe Aortic Valve Stenosis"

_diagnostics, 2023, doi:10.3390/diagnostics13111828_

Round 1
Reviewer 1 Report
I am an anatomist, and therefore can not thoroughly review the methods of this paper in that I can’t comment on whether or not there are other methods that should have been considered. However, the internal logic of the methods and the surprising results seem sound to me. The main comment I have to the authors is that this paper has very poor readability—there are just too many abbreviations and too much jargon in this paper to be understandable to the relatively generalist readership of Diagnostics, which includes many different subfield specialists. It took me an enormous amount of time to parse out the argument of this paper, and some abbreviations, such as NFHG, are never defined. The paper really needs to be redrafted to be more accessible to non-cardiologists, and the authors need to define more terms up front. In addition, the Introduction needs to include more context and background as to why this study is important and more information about the disease state of these patients. To name just one example, what is the difference between the effective orifice area and the geometric orifice area? Why is this difference important? What do these differences indicate about the health of the patients? Why are the results surprising, in that what is the underlying logic of the disease that led to these results? The authors are just expecting too much background knowledge on the part of the readers. I realize this is part of a special edition on structural heart disease, but there still needs to be some more context and definitions, as indicated by the special issue information.
Also, as a less important point, the current introduction needs to be proofed more carefully—for example, lines 70 and 71 are not a complete sentence, etc. etc.
In the end I think this will be a strong paper, with a rewrite that makes the paper more readable and understandable to the non-specialist.
Some minor reworking. Most of the problems in writing are not due to language difficulties, but to the excess use of jargon and abbreviations.
Author Response
I am an anatomist, and therefore cannot thoroughly review the methods of this paper in that I can’t comment on whether or not there are other methods that should have been considered. However, the internal logic of the methods and the surprising results seem sound to me. The main comment I have to the authors is that this paper has very poor readability - there are just too many abbreviations and too much jargon in this paper to be understandable to the relatively generalist readership of Diagnostics, which includes many different subfield specialists. It took me an enormous amount of time to parse out the argument of this paper, and some abbreviations, such as NFHG, are never defined. The paper really needs to be redrafted to be more accessible to non-cardiologists, and the authors need to define more terms up front. In addition, the Introduction needs to include more context and background as to why this study is important and more information about the disease state of these patients. To name just one example, what is the difference between the effective orifice area and the geometric orifice area? Why is this difference important? What do these differences indicate about the health of the patients? Why are the results surprising, in that what is the underlying logic of the disease that led to these results? The authors are just expecting too much background knowledge on the part of the readers. I realize this is part of a special edition on structural heart disease, but there still needs to be some more context and definitions, as indicated by the special issue information.
We really appreciate your important and very helpful comment. With respect to the mentioned aspects, we have added more background information and have tried to revise several paragraphs to improve the readability, even for non-cardiologists.
It took me an enormous amount of time to parse out the argument of this paper, and some abbreviations, such as NFHG, are never defined.
All abbreviations of AS subtypes are now defined at the end of this paragraph:
“Subsequently, severe AS patients were divided into four subgroups: LFLG (low flow-low gradient)-AS, NFLG (normal flow-low gradient)-AS, LFHG (low flow-high gradient)-AS, and NFHG (normal flow-high gradient)-AS.”
Page: 4, Line: 137-139
Questions: “What is the difference between the effective orifice area and the geometric orifice area?” “Why is this difference important?" “What do these differences indicate about the health of the patients?”
In the introduction, more background information about the effective and geometric orifice area is now given and differences between the two parameters are described:
“Nevertheless, the EOA or geometric orifice area (GOA) represents the actual target value for the assessment of AS severity and thus forms the basis for further divisions into hemodynamic AS subgroups [4,5].
The EOA describes the functional orifice area, which can be calculated indirectly by the continuity equation. When determining EOA, some methodological limitations must be considered. For example, in comparison to the GOA, the EOA is suspected to be smaller, which is due to the phenomenon of flow constriction caused by flow turbulence at the inner edges of the actual anatomic orifice area [6]. In contrast, the GOA corresponds to the anatomic orifice area, which can directly be measured by echocardiography. Semi-invasive transesophageal (3D) echocardiography (TEE) enables a more precise assessment of GOA and aortic dimensions. TEE is also recommended when components of the continuity equation, e.g. the left ventricular outflow tract (LVOT) diameter, cannot be determined precisely by TTE. Further, LV volumes and function, as well as aortic dimensions can be determined more precisely by 3D echocardiography[7–9].”
Page: 2, Line: 50-64
Question: “Why are the results surprising, in that what is the underlying logic of the disease that led to these results?”
The fundamental difference between an isolated AS and an AS with concomitant AR is that the presence of AR leads to a fundamental difference in the total and effective stroke volume. This fundamental statement is now stated in the following paragraph of the Introduction:
“The increase of forward blood flow through the aortic valve in the presence of AR will not influence the calculation of EOA. However, the flow conditions, which are estimated by forward stroke volume through the aortic valve determined by Doppler echocardiography, do not reflect effective stroke volume. This difference between total and effective stroke volume in mixed aortic valve disease might influence the classification of patients with different AS subgroups.“
Page: 2, Line: 68-73
Also, as a less important point, the current introduction needs to be proofed more carefully - for example, lines 70 and 71 are not a complete sentence, etc. etc.
Thank you for pointing this out. Both passages have been corrected accordingly.
Page: 2, Line: 88
Reviewer 2 Report
I am grateful for the opportunity to review this paper.
It is well known that the EOA is the most reliable indicator for severity classification of AS, rather than flow velocity and pressure gradient. This is because flow velocity and pressure gradient are affected by many things, including AR.
I have a few comments.
1. EAO should be corrected to EOA, line 216.
2. Did you perform TEE on all patients to measure GOA? If not, what are the criteria for performing TEE? 3. As stated in the Limitation, why did you use PHT for AR severity classification? We believe that a
quantitative assessment such as EROA is a more reliable indicator.
Author Response
I am grateful for the opportunity to review this paper. It is well known that the EOA is the most reliable indicator for severity classification of AS, rather than flow velocity and pressure gradient. This is because flow velocity and pressure gradient are affected by many things, including AR.
I have a few comments.
- EAO should be corrected to EOA, line 216.
Thank you very much for pointing out this mistake, which has now been corrected.
Page: 8, Line: 238
- Did you perform TEE on all patients to measure GOA? If not, what are the criteria for performing TEE?
Thank you for this important comment. In all patients TTE and TEE was performed to measure EOA as well as GOA.
Page: 2, Line: 82-84
- As stated in the Limitation, why did you use PHT for AR severity classification? We believe that a quantitative assessment such as EROA is a more reliable indicator.
Thank you very much, we really appreciate this important comment.
The semi-quantitative assessment of AR severity by EROA using the proximal isovelocity surface area (PISA) method can be seen as a more reliable indicator. However, it needs to be noted, that in patients with severe AS and only trace or mild AR (majority of the patients in the present study), the flow convergence zones (PISA zones) can hardly be visualized. Further, due to the fact, that this is a retrospective study the flow convergence zone could not be elaborated well in a high number of patients with primarily trace and mild AR. On the other hand, PHT is a semi-quantitative parameter which is also mentioned in current recommendations, easy to assess and has been routinely assessed in all patients with concomitant AR.
The following paragraph has been added in the limitations:
“Although the effective regurgitant orifice area (EROA) determined by the proximal isovelocity surface area (PISA) method may be considered a more reliable parameter, AR severity was assessed semi-quantitatively by PHT in the present study. It should be noted that in patients with severe AS and only trace or mild AR (most patients in the present study), flow convergence zones (PISA zones) are difficult to visualize, and the PISA method is often hardly applicable. Because this is a retrospective study, the flow convergence zones could not be well elaborated in many patients with primarily trace and mild AR. On the other hand, the PHT method is also mentioned in current recommendations, despite its main limitation that alignment of AR jet formation by cw-Doppler is often difficult.“
Page: 10, Line: 329-338
Reviewer 3 Report
I read with great interest the manuscript by Kanders et al. on the impact of AR on doppler echocardiographic parameters in patients with severe AS. They analyzed TTE and TEE data on 238 patients retrospectively, thus finding a significant impact on the Doppler derived echocardiographic parameters maxVAV and mPGAV, whereas EOA and the ratio of maxVLVOT/maxVAV remain unchanged.
The paper is original and well written. I have only some minor issues to be addressed.
Introduction
- Line 44-45. When you introduce the value of transthoracic echocardiography, you should also underline the role of TTE in evaluating systolic function (doi: 10.1186/s12947-017-0121-8), diastolic function (doi: 10.1111/echo.15462), and fluid responsiveness (doi: 10.1016/j.jcrc.2022.154108). Please discuss and add these 3 references.
Discussion
- Line 218: Please explain why you expected significant difference between both parameters in patients with isolated severe AS.
- Line 257: Difference is misspelled
Limitation
- Please move the limitation section after the discussion section but before the conclusions. Please also add a brief "strenght" paragraph highlighting the strong points of the article, such as the evaluation of inter- and intra-observer variability.
I have no comments
Author Response
I read with great interest the manuscript by Kandels et al. on the impact of AR on doppler echocardiographic parameters in patients with severe AS. They analyzed TTE and TEE data on 238 patients retrospectively, thus finding a significant impact on the Doppler derived echocardiographic parameters maxVAV and mPGAV, whereas EOA and the ratio of maxVLVOT/maxVAV remain unchanged. The paper is original and well written. I have only some minor issues to be addressed.
Introduction
- Line 44-45. When you introduce the value of transthoracic echocardiography, you should also underline the role of TTE in evaluating systolic function (doi: 10.1186/s12947-017-0121-8), diastolic function (doi: 10.1111/echo.15462), and fluid responsiveness (doi: 10.1016/j.jcrc.2022.154108). Please discuss and add these 3 references.
We really appreciate this interesting and helpful comment and have read these references with great interest.
10.1186/s12947-017-0121-8 (systolic function)
The following reference has been added to the present study.
Page: 2, Line: 63-64 (reference no. 9)
10.1111/echo.15462 (diastolic function)
This is a very interesting manuscript assessing echocardiographic parameters of diastolic function in COVID-19 patients admitted to ICU. The cohort of ICU patients with COVID 19 significantly differs from the cohort in our study and could unsettle the reader. We believe that this citation is not directly linked to the topic of the present manuscript. Nevertheless, the current recommendation to analyze diastolic function by echocardiography (Nagueh et al. 2016) has been added to the references including the following paragraph in the methods:
“Diastolic function was assessed by E/E’, E/A and systolic pulmonary artery pressure (sPAP) according to current recommendations [19].”
Page: 4, Line: 110-112 (reference no. 20)
10.1016/j.jcrc.2022.154108 (fluid responsiveness)
This is also a very interesting manuscript, but to our opinion this reference might also unsettle the reader. Again, we believe that this reference is not directly linked to the topic of the present manuscript. However, the echocardiographic assessment of volume state and fluid responsiveness is also mentioned in the current recommendations by Nagueh et al. (2016).
Discussion
- Line 218: Please explain why you expected significant difference between both parameters in patients with isolated severe AS.
Thank you very much for this important comment. This sentence was misleading and is now rewritten:
“As suspected, there was no significant difference between both parameters in patients with isolated severe AS.”
Page: 8, Line: 238-239
- Line 257: Difference is misspelled
Thank you very much for pointing out this mistake, which has now been corrected.
Page: 9, Line: 278
Limitation
- Please move the limitation section after the discussion section but before the conclusions. Please also add a brief "strength" paragraph highlighting the strong points of the article, such as the evaluation of inter- and intra-observer variability.
We really thank the reviewer for this helpful comment.
A paragraph “Strengths and limitations” has been added before the conclusion. The following Strengths have been highlighted:
„In the present study a relatively high number of well-characterized patients with severe AS according to continuity equation were included (n=283). In addition, all patients underwent TEE to confirm the diagnosis. Low intra- and interobserver variabilities highlight the quality of the acquired and analyzed echocardiographic data sets. In addition to the echocardiographic data, data on the patients' symptoms were also available.”
Page: 10, Line: 323-327
Reviewer 4 Report
In this manuscript, the authors describe the impact of concomitant AR on doppler echocardiographic parameters in patients with severe AS. I think this manuscript is well written and it is worthwhile to be published in this journal with some revision. I have a couple of suggestions.
- In discussion, if the authors could add comments on invasive pressure measurement (left heart catheterization) as an alternative exam, it would be better because the patients with severe AS usually need left heart catheterization and also TEE is an invasive test.
- Page 10, Line 216: "EAO" should be "EOA".
Author Response
In this manuscript, the authors describe the impact of concomitant AR on doppler echocardiographic parameters in patients with severe AS. I think this manuscript is well written and it is worthwhile to be published in this journal with some revision. I have a couple of suggestions.
- In discussion, if the authors could add comments on invasive pressure measurement (left heart catheterization) as an alternative exam, it would be better because the patients with severe AS usually need left heart catheterization and also TEE is an invasive test.
We would like to thank the reviewer for this helpful suggestion.
The following paragraph has been added in the discussion:
“Invasive measurements of pressure gradients to assess AS severity are possible in principle but are only recommended in patients where non-invasive cardiac imaging is inconclusive or discordant with clinical findings. Nevertheless, in patients with severe AS and high cardiovascular risk profile, coronary heart disease should be excluded by coronary angiography before further treatments, e.g., transcatheter aortic valve implantation or surgery, will be performed [3].”
Page: 9, Line: 286-291
- Page 10, Line 216: "EAO" should be "EOA".
Thank you very much for pointing out this mistake, which has now been corrected.
Page: 8, Line: 238
Round 2
Reviewer 1 Report
I think it can now be accepted. Great job on the rewrite.
Reviewer 2 Report
It is my great pleasure to be re-invited to review this submission. Authors revised their manuscript extensively based on reviewers' comments and the manuscript was improved. I have no more comments or questions.